# Discrete energy levels of Caroli-de Gennes-Matricon states in quantum limit in FeTe$_{0.55}$Se$_{0.45}$

Mingyang Chen[1], Xiaoyu Chen[1], Huan Yang[1], Zengyi Du[1], Xiyu Zhu[1], Enyu Wang[1] & Hai-Hu Wen[1]

Caroli-de Gennes-Matricon (CdGM) states were predicted in 1964 as low-energy excitations within vortex cores of type-II superconductors. In the quantum limit, the energy levels of these states were predicted to be discrete with the basic levels at $\pm\mu\Delta^2/E_F$ ($\mu = 1/2, 3/2, 5/2,$ ...) with $\Delta$ the superconducting energy gap and $E_F$ the Fermi energy. However, due to the small ratio of $\Delta/E_F$ in most type-II superconductors, it is very difficult to observe the discrete CdGM states, but rather a symmetric peak which appears at zero bias at the vortex center. Here we report the clear observation of these discrete energy levels of CdGM states in FeTe$_{0.55}$Se$_{0.45}$. The rather stable energies of these bound state peaks vs. space clearly validate our conclusion. Analysis based on the energies of these CdGM states indicates that the Fermi energy in the present system is very small.

[1] Center for Superconducting Physics and Materials, National Laboratory of Solid State Microstructures and Department of Physics, Collaborative Innovation Center for Advanced Microstructures, Nanjing University, 210093 Nanjing, China. These authors contributed equally: Mingyang Chen, Xiaoyu Chen. Correspondence and requests for materials should be addressed to H.Y. (email: huanyang@nju.edu.cn) or to H.-H.W. (email: hhwen@nju.edu.cn)

When magnetic field is applied to a type-II superconductor, vortex with quantized flux of $\Phi_0 = h/2e = 2.07 \times 10^{-15}$ Wb will be formed. At the center of a single vortex, the superconducting order parameter is zero and it recovers a unified value in the scale of coherence length $\xi$. Due to the confinement to the quasiparticles by the vortex core, Caroli-de Gennes-Matricon (CdGM)[1] predicted that there are confined low-energy bound states with the energy levels at about $E_\mu = \pm \mu\Delta^2/E_F$ ($\mu = 1/2, 3/2, 5/2, \ldots$) with $\Delta$ the superconducting energy gap and $E_F$ the Fermi energy. However, due to the very-small value of $\Delta/E_F$, these discrete energy levels of CdGM states have never been really observed. In most conventional superconductors, the easily observed feature is that a peak of density of states (DOS) locates at zero energy with a symmetric shape[2,3]. This peak will split and fan out when moving away from the center. Later on, it was understood that[4–6] this symmetrized peak around zero energy at the vortex center is due to the accumulated DOS arising from many symmetric CdGM states when the quantum limit situation $T/T_c \ll \Delta/E_F$ is not satisfied. The theoretical calculations starting from the Bogoliubov-de Gennes equations can explain not only the symmetrically shaped bound state peak at Fermi level at the core center, but also the splitting into two peaks that exhibit the fanning out behavior at positions away from the center. Although the bound state peak of the lowest energy level has been argued for the quantum limit in YNi$_2$B$_2$C (ref.[7]), YBa$_2$C$_3$O$_{7-\delta}$ (ref.[8]), Bi$_2$Sr$_2$CaCu$_2$O$_{8+\delta}$ (ref.[9]), and iron-based superconductors[10–12], while the clear evidence of the discrete CdGM states is still lacking. For example, the vortex core states exhibit as a dominant asymmetric peak near Fermi level in YNi$_2$B$_2$C (ref.[7]) and iron-based superconductors[10–12]. In cuprate superconductors, one pair of roughly symmetric peaks on scanning tunneling spectrum were observed within the vortex core, but it was argued that these may result from the competing orders[8,9,13].

In this study, we report the measurements of the local DOS crossing the vortex in the iron-based superconductor FeTe$_{0.55}$Se$_{0.45}$. We have observed the discrete vortex core states near zero bias, which suggests the observation of the discrete CdGM states in the quantum limit.

## Results

**Magnetization, topography, and spectroscopy studies on FeTe$_{0.55}$Se$_{0.45}$.** Figure 1b shows the temperature dependence of the mass magnetization of the FeTe$_{1-x}$Se$_x$ ($x = 0.45$, nominal composition) single crystal, and the superconducting transition occurs at about $T_c = 13.3$ K. Figure 1a shows a typical topographic image measured by scanning tunneling microscopy (STM). One can observe clear atomically resolved topography with the lattice constant of $3.79 \pm 0.02$ Å in two perpendicular lattice directions, which is consistent with 3.8 Å of other measurements[14,15]. According to previous studies[14–17], the brighter spots on the surface represent the Te atoms, while the darker spots are the Se atoms with smaller atomic size. When we do the counting on the numbers of Te and Se atoms in Fig. 1a, the

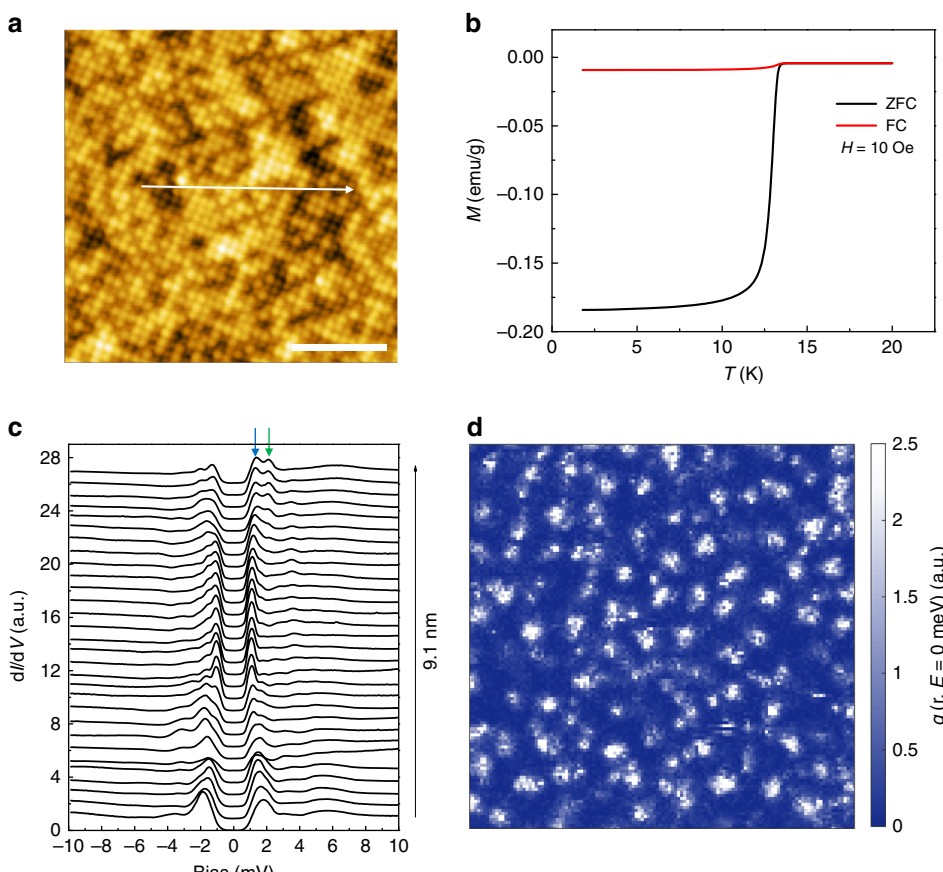

**Fig. 1** STM results and superconducting transition of FeTe$_{0.55}$Se$_{0.45}$. **a** Atomically resolved topographic image with the square lattice measured at 1.8 K with a bias voltage of $V_{bias} = 10$ mV and tunneling current of $I_{set} = 300$ pA. Scale bar, 4 nm. **b** Temperature dependence of mass magnetization after zero-field-cooled (ZFC) and field-cooled (FC) modes at 10 Oe. **c** Spatially resolved tunneling spectra along the arrowed line measured at 0.4 K. The superconducting gaps seem very inhomogeneous on the sample and two arrows on one typical spectrum are used to mark the two energy gaps. **d** Spectroscopic image of vortex lattice measured by zero bias conductance map at $T = 0.48$ K and $B = 5$ T, the field of view dimensions are 200 nm × 200 nm

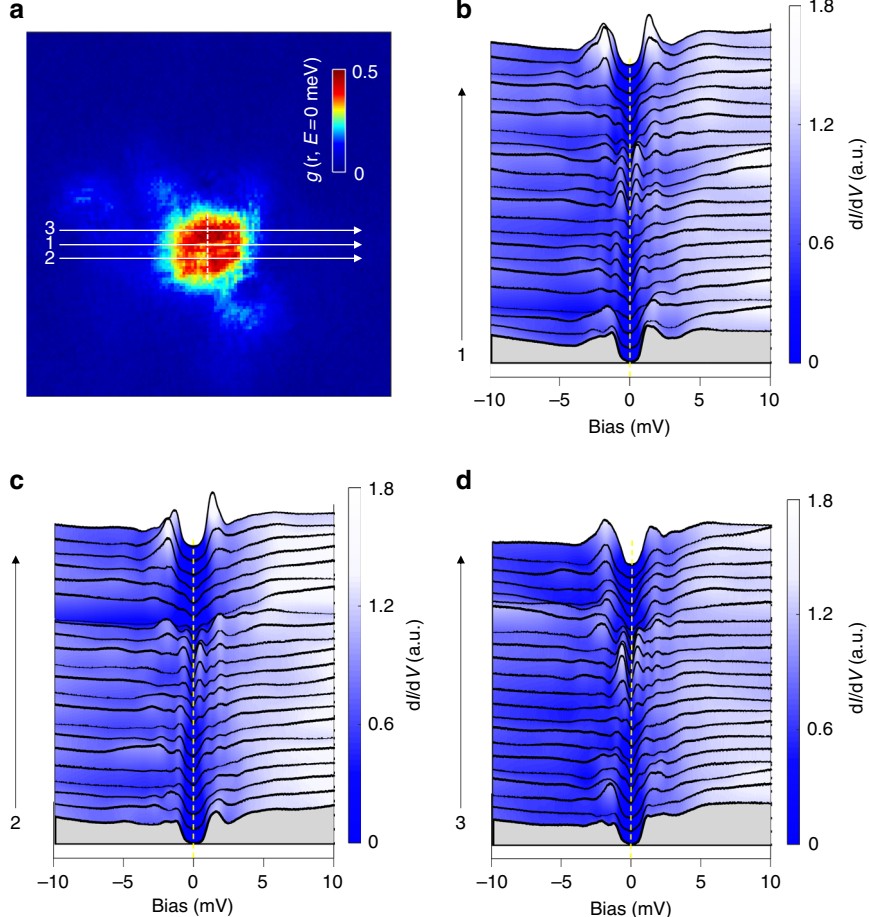

**Fig. 2** Vortex image and CdGM states near the vortex core center. **a** Image of a single vortex in a 20 nm × 20 nm region measured at 0.48 K and 4 T. **b–d** Tunneling spectra measured along the arrowed lines marked from 1 to 3 in **a** with increment steps of 7.6 Å. The dashed lines in **b–d** show the positions of zero bias voltage for each figure. The discrete CdGM bound state peaks can be clearly observed near the vortex core center

resultant composition is roughly $FeTe_{0.54}Se_{0.46}$, which is very close to the nominal formula $FeTe_{0.55}Se_{0.45}$. No interstitial Fe atoms with signatures of large bright spots[16] have been observed on the cleaved top surface in our measurements, which manifests that most of the interstitial Fe impurities have been removed by the annealing treatment (see Methods). Figure 1c shows a series of scanning tunneling spectroscopy (STS) measured along the arrowed line in Fig. 1a at $T = 0.4$ K. Obviously our measured spectra show a fully gapped feature everywhere with one or two pairs of superconducting coherence peaks, which is consistent with previous reports[14,16]. The statistics on the peak positions reflecting the superconducting gaps on the same sample is shown in Supplementary Fig. 1. On a specific tunneling spectrum, we can see one or two pairs of coherence peaks with energies varying from 1.1 to 2.1 meV. The two-gap feature observed on one tunneling spectrum indicates a multiband nature, and the particular shape of the spectrum, especially the part near the coherence peaks, strongly depends on the local details of the surface.

**Vortex image and discrete vortex bound state peaks**. When a magnetic field of 5 T is applied, vortices can be observed on the sample by measuring the zero bias conductance map as shown in Fig. 1d. One can see that the vortex lattice is disordered and is obviously very different from the ordered vortex lattice observed on $Ba_{0.6}K_{0.4}Fe_2As_2$ (ref. [10]). The disordered vortices are also observed in $FeTe_{1-x}Se_x$ in previous study[12]. These vortices are pinned by some regions of weak superconductivity or local

defects, and it is consistent with the inhomogeneous electronic properties reflected by the spatial inhomogeneity of the tunneling spectra shown in Fig. 1c. If we enlarge the view on an individual vortex, we can find that many of them have anisotropic or irregular shapes in real space. This seems to be a common feature in both cuprate and iron-based superconductors. Although we did not observe the clear evidence of checkerboard electronic state within the vortex core as observed in $Bi_2Sr_2CaCu_2O_{8+\delta}$ (ref.[18]) and $Ca_{2-x}Na_xCuO_2Cl_2$ (ref.[19]), the shape of an individual vortex in our sample is strongly affected by the inhomogeneity of surface electronic structure, this manifests that the superconductivity may be unconventional.

In order to study the vortex core state, we measured the tunneling spectra across a single vortex. In Fig. 2a, we show the dI/dV mapping measured at zero bias. One can clearly see a single vortex with a roughly round shape. Following the parallel traces marked by the three white arrowed lines in Fig. 2a, we measured the tunneling spectra across the vortex with increment steps of 7.6 Å, and show the results in Fig. 2b–d, respectively. It is worth mentioning that the number of spectra in this figure has been diluted into half for clarity; the original data with twice density of measured spectra along the same lines are shown in Supplementary Fig. 2. One can see that the tunneling spectra measured far away from the vortex center are similar to those measured at 0 T. Near the center of the vortex, the spectra reveal two remarkable features. Firstly, the superconducting coherence peaks are suppressed completely, indicating a suppression of superconducting order parameter in this region. Secondly, there are some

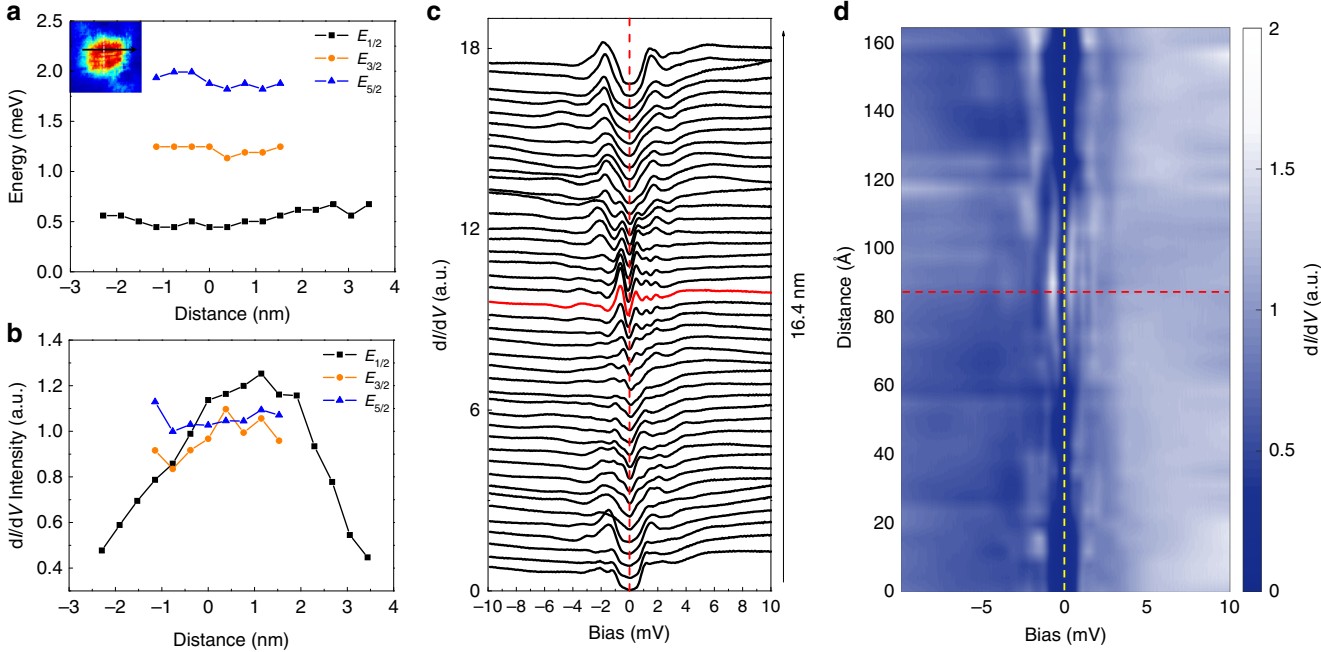

**Fig. 3** Vortex bound state peak positions and intensities. **a** Peak energies of $E_{1/2}$, $E_{3/2}$, and $E_{5/2}$ as function of tip position along the trace marked by the black line crossing a vortex shown in the inset. **b** The differential conductance intensity of bound state peaks at the peak energies of $E_{1/2}$, $E_{3/2}$, and $E_{5/2}$ as the values in **a**. **c** Spatially resolved tunneling spectra measured across the vortex and the red curve represents the spectrum measured at the center of the vortex core. Each curve is offset for an increment distance of 3.8 Å. **d** Color plot of spatial profile of the spectra crossing the vortex. The horizontal red dashed line represents the position of the vortex core center. The vertical yellow dashed line marks the zero bias

discrete bound state peaks, and those close to the zero bias exhibit roughly a symmetric shape. The peaks near the zero bias locate actually at $+0.5 \pm 0.05$ meV and $-0.6 \pm 0.05$ meV, respectively. The slight difference of the peak energies at positive and negative bias may be induced by the local impurities or disorders[20]. The vortex core states in FeTe$_{0.55}$Se$_{0.45}$ observed here are apparently different from those of conventional superconductors, e.g., 2H-NbSe$_2$ (ref.[2]) in which a giant peak was observed symmetrically around zero bias and it fans out when moving away from the vortex center. The feature of discrete bound state peaks vanishes near the edge of the vortex core. We argue that these peaks are exactly the vortex bound states in the quantum limit. When we focus on the bound state peaks at positive bias side, there are three discrete peaks between the zero bias and the larger superconducting gap. If we attribute the vortex core states to CdGM type in the quantum limit, the three peaks locating at $+0.45$, $+1.2$, and $+1.9$ meV may refer to the low-lying bound states with $\mu = 1/2$, 3/2, and 5/2, respectively. Actually the three peak energies give a ratio of 1:2.7:4.2, which is not so far from the ideal case of 1:3:5 of the first, second, and third energy levels of the CdGM bound states. Considering the first energy level at positive bias $E_{1/2} = \Delta^2/2E_F \approx 0.45$ meV, combining with $\Delta$ from 1.1 to 2.1 meV, we can obtain the Fermi energy $E_F$, which ranges from 1.3 to 4.9 meV. The calculated value of Fermi energy is really very small and comparable to the superconducting gap. The small Fermi energies determined here are very close to those obtained from angle-resolved photoemission spectroscopy (ARPES) measurements[21,22], which yield a Fermi energy as low as 4 meV. The measurements along three vertical traces are also done for the same vortex, and the results are similar and shown in Supplementary Fig. 3.

To further elaborate the properties of the discrete energy levels in the vortex core, we extract the energy values and intensities of the three discrete peaks at positive bias and show them in Fig. 3a, b as a function of distance along the trace marked by the dark

arrowed line in the inset of Fig. 3a. The spectra with offset for increment spatial step of 3.8 Å are shown in Fig. 3c. It is clear that the peak energies of $E_\mu$ ($\mu = 1/2$, 3/2, and 5/2) are almost unchanged with the STM tip position, but the intensity of $E_{1/2}$ peaks decreases when the tip moves away from the vortex center. The feature is better shown in Fig. 3d by a two-dimensional color plot of the spatial evolution of the tunneling spectra. It is obvious that the peaks of the discrete energy levels near the vortex center are almost independent of positions yielding three parallel ridge-like traces in the positive bias side. Such tendency remains until the STM tip moves out of the vortex core region. The characteristics of these bound states are consistent with the CdGM states in quantum limit by theoretical predictions[23]. If we plot the spatially dependent amplitude of $dI/dV$ taken at bias of $E_{1/2} = 0.45$ mV for this vortex along the six different traces in a wider region, it is easy to see a dip at around $+3.82$ nm and $-3.44$ nm. Outside the dip, a pair of small second peaks of $dI/dV$ amplitude appears at about $\pm 4.5$ nm away from the vortex center for most line-cuts on this vortex. The related results are shown in Supplementary Fig. 4a. We argue that this dip and second peak may be the theoretically predicted amplitude oscillation of CdGM states in quantum limit[23], which thus provides another proof and is further discussed in Supplementary Note 1.

Focusing on the spectrum near the vortex core center, as shown by the red line in Fig. 3c, one can see that the spectrum is strongly particle-hole asymmetric, which manifests in two ways. First, three bound state peaks appear in the positive energy side, but only one in the negative energy side. Second, the differential conductance intensity of the lowest bound state peak on negative energy is slightly higher than that on positive energy. This asymmetric feature seems to be different from the theoretical results for single-band system[23] and the experimental results in YNi$_2$B$_2$C (ref.[7]), in both cases the $E_{1/2}$ peak is much stronger than $E_{-1/2}$ peak. We believe that this asymmetric spectrum and different features on the positive and negative energy side are

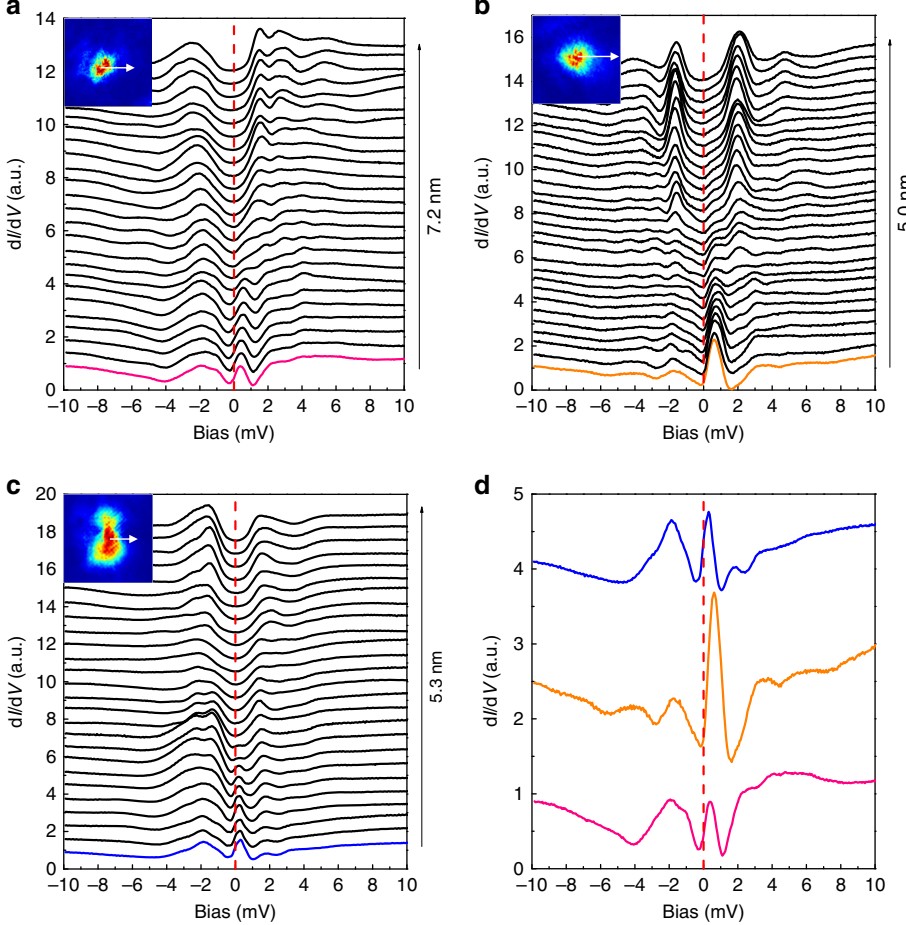

**Fig. 4** Vortex core with single bound state peak. **a–c** Series of spatially resolved tunneling spectra measured at 0.48 K and 4 T along the traces marked by the white lines drawn in the insets, respectively. The pink, orange, or blue curve in **a–c** represents the spectrum measured at different vortex centers. **d** Spectra measured at the center of three different vortices. The red dashed lines in all figures show the positions of zero bias voltage. For these three vortices, there is only one bound state peak appearing at a positive bias near the vortex center

induced by the existence of shallow electron and hole bands with different Fermi energies in the system. ARPES measurements[21] show that the two hole bands ($\alpha_2$ and $\alpha_3$) with superconducting gaps of about 2 meV cross the Fermi level with $E_F \approx 4$ meV, and the electron band has a bottom energy of about 10 meV from the Fermi level with a superconducting gap of about 5 meV. Since our measurements reveal mainly the maximum gap of about 2 meV, we attribute the CdGM bound states mainly to the collective contributions of the two hole bands. Due to the shallow hole bands, the bound state intensity and distributions are naturally asymmetric with respect to zero bias. One may argue that the second and third peaks on the spectrum at the vortex core center have similar energies of the superconducting gaps far away from the vortex core, thus they could be the residual gap feature of the system. This is unlikely and not possible. According to the basic understanding of a vortex structure based on the Ginzburg–Landau theory, all pairing order parameter at the vortex core center should vanish, this leaves no space for a residual gap feature with the unchanged gap values.

**Other kinds of vortex bound states**. In addition to the discrete energy levels mentioned above, we also observed some vortices with different patterns of vortex bound states. In some cases, we find the tunneling spectra showing one dominant, but extremely asymmetric bound state peak locating near zero bias. Figure 4

shows three sets of tunneling spectra crossing different vortices observed on different samples with this kind of feature. This type of tunneling spectrum is consistent with previous observation within the vortex core of iron-based superconductors[10–12]. Although the images of three vortices shown in the insets of Fig. 4a–c have slightly different shapes, the main features of these vortex bound states seem to be similar and are very different from those shown in Figs 2 and 3. From the spatially resolved tunneling spectra in Fig. 4a–c, one can see that the peak positions of the vortex bound states locate at some positive bias voltage from 0.2 to 0.6 meV. However, all the peaks do not split or change their peak positions when the tip moves away from the vortex center, which is similar to the situation of discrete bound states. The extremely asymmetric feature of the vortex bound states in these vortices may be induced by the local details of electronic structure. Theoretically, it was predicted that the peak with $\mu = 1/2$ may have very-high intensity in the quantum limit in a single-band system[23]. The asymmetry of the DOS in the occupied and unoccupied states in iron-based superconductors due to multiple shallow bands may also lend an explanation for the extremely asymmetric bound state peak intensity at positive and negative bias in our present system[15,24]. Since vortices favor to be pinned by defects or disorders, which may give strong influence on the detailed shape of bound state peaks in the present sample. Statistically, we have conducted measurements on more than 30 vortices, at least 20% of them show the discrete energy levels,

most of the rest show the asymmetric single peak at a positive energy and near zero bias, but none of them shows the peak position exactly at zero. In Supplementary Fig. 5, we present the spatial evolution of tunneling spectra crossing other three vortices with discrete bound state peaks beside those shown in Fig. 2. The vortex core states are strongly influenced by the local defect or the surrounding vortices[25], or the asymmetric density of state to Fermi level in the normal state[26]. These may give explanations for different patterns of bound state peaks in different vortices. This will be further addressed in next section.

From our experiment, we observe different kinds of CdGM bound states in different vortices. For the purpose of control experiment, we have repeated to observe three bound state peaks on the positive bias in more vortices in different samples, and the results are shown in Supplementary Fig. 6a, b, d, e. Although the oscillating amplitude is not as strong as that presented in Fig.2, the general features look similar. We fully believe they have the same origin. In order to understand what causes the different vortex bound state patterns, we have also measured the bare spectra at zero field in the same areas of the vortices. For these newly observed vortices with the three bound state peaks, the spatial evolution of spectra at zero field are shown in Supplementary Fig. 6c, f. One can see that the spectra look quite uniform. For the vortices with one dominant vortex bound state peak, we have intentionally selected one vortex and measured the spatial evolution of spectra at zero field in the area, where one such vortex would appear when a magnetic field is applied, which is shown in Supplementary Fig. 7. It is easy to see that in the core region the gap becomes clearly smaller than that in the region far away from the core. Therefore, it is tempting to conclude that the defect induced pinning can modify the structure of the vortex core bound states and may smear up the peaks of different quantum levels. In addition, we have a feeling that, for the vortices having the discrete and higher order energy levels of the CdGM bound state peaks, the vortex image usually looks round in shape. While those with one dominant peak seem to have irregular shapes, which may suggest an influence of local defect or impurities on the vortex core bound state pattern.

## Discussion

As presented above, we have observed the discrete CdGM states with higher order peaks ($\mu > 1/2$). One may argue that these higher order peaks are induced by some competing orders or impurity states of defects. However, in our point of view, if the competing orders or impurity states exist, it is difficult to imagine why we should have three characteristic energy scales with the ratio of being close to the ideal case 1:3:5 of the CdGM bound states. Furthermore, in iron-based superconductors, the possible competing orders include the nematic phase and the anti-ferromagnetic order. In FeTe$_{1-x}$Se$_x$ on the doping level of $x = 0.45$, it is known that these orders are absent. Given that these competing phases may emerge in the vortex core, but the energy scales should be much larger than those observed here. In addition, the rather uniform tunneling spectra crossing the same area of the vortex when the magnetic field is zero give strong objection to the picture of defect or impurity.

On all the spectra measured on different vortices, we have not observed any bound state peak, which appears exactly at zero bias. The offset bias in the tunneling spectrum measurements may affect the exact peak positions; thus, we have done careful calibration on the offset bias (see Supplementary Note 2). For statistics, we collect the energies of bound state peaks close to the zero bias from all measured nine vortices presented in this paper. The results are shown in Supplementary Fig. 9. All the lowest

bound state peaks locate at energies from about 0.2 to 0.8 meV for both positive and negative bias.

For both types of the vortex bound state pattern we witnessed here, the peak energies almost do not shift when the STM tip moves away from vortex center. This is consistent with the theory for the CdGM bound states in the quantum limit; therefore, we attribute them all to the CdGM states. In five vortices, we observed the discrete peaks of CdGM states. In conventional superconductors with large $E_F$, the giant bound state peak measured at the core center will fan out and move to larger bias voltage when the tip moves away from the vortex center. However, the vortex bound state peak energy in quantum limit will not shift by moving the STM tip on the superconductors with very-small $E_F$. By using the Pippard relation $\xi_0 = \hbar v_F / \pi \Delta$, it is easy to derive that $\Delta / E_F \sim 1/k_F \xi_0$. As mentioned before, in the quantum limit we have $T/T_c \ll \Delta / E_F$. In FeTe$_{1-x}$Se$_x$ samples, $k_F$ (or $E_F$) is very small as revealed by ARPES for both hole and electron pockets[21,22]. We did the measurements at about 400 mK, and $T/T_c \approx 0.03$, which is much smaller than $1/k_F \xi_0 \approx 0.3$ to 0.6 [using $\xi_0 = 25$ Å, $k_F$ ($\alpha$-band) $= 0.07$ Å$^{-1}$, and $k_F$ ($\beta$-band) $= 0.12$ Å$^{-1}$ (refs[21,22,27])], with $\xi_0$ the coherence length in zero-temperature limit. Therefore, this analysis gives strong argument for the existence of discrete CdGM states in quantum limit in present system. As we mentioned already, none of our tunneling spectra exhibits the bound state peak exactly at zero, this is different from a recent observation, which claims the possible evidence of Majorana mode[28]. We do not want to exclude that possibility since only partial bands are involved in forming the possible Dirac cone structure[29] in the material, and perhaps that experiment just successfully detects the surface states of that band. Our present experiments provide clear evidence for the discrete CdGM bound state peaks, which has been long sought for decades.

## Methods

**Sample synthesis**. The FeTe$_{1-x}$Se$_x$ single crystals with nominal composition of $x = 0.45$ were grown by self-flux method[30]. The excess Fe atoms were eliminated by annealing the sample at 400 °C for 20 h in O$_2$ atmosphere followed by quenching in liquid nitrogen.

**STM/STS measurements**. The STM/STS measurements were carried out in a scanning tunneling microscope (USM-1300, Unisoku Co., Ltd.) with ultra-high vacuum, low temperature, and high magnetic field. The samples were cleaved in an ultra-high vacuum with a base pressure about $1 \times 10^{-10}$ torr. Tips made of Pt/Ir alloy or W were used for the STM/STS measurements. A typical lock-in technique was used for the tunneling spectrum measurements with an AC modulation of 0.3 mV and 973.8 Hz. Most of the tunneling spectra were recorded in the tunneling condition of $V_{bias} = 10$ mV and $I_{set} = 500$ pA. The offset bias voltages in STS measurements have been carefully calibrated. The vortex images are measured by zero bias conductance mapping. Since the distance between the tip and the sample is different in different set of STM measurements, the spectra should be normalized in order to have a valid comparison. For most spectra, the normalizing factor is 10, except for those in Fig.4b it is 3; in Supplementary Fig. 6a and d it is 1; in Supplementary Fig. 6c and f it is 2.

**Data availability**. The data that support the plots within this paper and other findings of this study are available from the corresponding author upon reasonable request.

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

## Acknowledgements

We acknowledge useful discussions with Tao Xiang, Jenny Hoffman, Tetsuo Hanaguri, Christoph Berthod, and Dimitri Roditchev. The work was supported by National Key R&D Program of China (grant number: 2016YFA0300401), National Natural Science Foundation of China (NSFC) with the projects: 11534005, 11374144, and Natural Science Foundation of Jiangsu (grant number: BK20140015).

## Author contribution

The low-temperature STM/STS measurements and analysis were performed by M.Y.C, X.Y.C, H.Y., and H.H.W. The samples were grown by Z.Y.D and X.Y.Z. H.Y., M.Y.C., and H.H.W contributed to the writing of the paper. H.Y. and H.H.W. are responsible for the final text. All authors have discussed the results and the interpretations.

## Additional information

**Competing interests:** The authors declare no competing interests.

