## [Peer Review File · Nature Communications]

Reviewers' comments:

Reviewer #1 (Remarks to the Author):

The manuscript presents new detailed low temperature STM data revealing the existence of Caroli-de Gennes-Matricon states in tunneling spectra in FeTe_{0.55}Se_{0.45} superconductors. These data shed new light on the nature of quantum states in the vortex core in superconductors and will be certainly of great interest to the readers of Nature Communications and other Journals.

The data are in general well presented and the manuscript can be accepted for publication in Nature Communications, provided that the authors revise the part describing the asymmetry of the CdGM peaks in tunneling spectra. More specifically, more convincing answers should be given concerning the possible nature of the asymmetry of the observed CdGM peaks with respect to zero bias.

Reviewer #2 (Remarks to the Author):

Chen et al. report the observation of peaks in the tunneling conductance inside superconducting vortices in FeTeSe using scanning tunneling microscopy. They argue that the structure of these peaks both in space and energy is consistent with them being the result of discrete Caroli-de Gennes-Matricon (CdGM) states. Furthermore, they conclude that the observation of discrete CdGM states means that FeTeSe has a small Fermi energy.

Novelty, Significance and Interest:

The main novel claim of this manuscript is the observation of discrete CdGM states in superconducting vortices, an experimentally demanding task. The observation of discrete CdGM states implies a small Fermi energy in FeSeTe. This has previously been inferred using ARPES in refs. 21 and 22 and so will not significantly influence thinking in the field. Aside from this the measurements presented do not shed light on the physics of FeTeSe and related compounds. Consequently, it is mainly of interest to STM experimentalists and theorists who work on vortex core structure.

The authors present measurements of vortices exhibiting sub-gap states that fall into two categories a) discrete peaks symmetrically positioned around zero bias b) single peak slightly offset from zero bias. As stated in the manuscript, these two phenomena have previously been observed (albeit not together) in cuprates (refs. 8,9,13), YNi₂B₂C (ref. 7) and other iron-based superconductors (refs. 10-12).

One of the reported vortices exhibits multiple sub-gap peaks with spatial structure consistent with CdGM states in the quantum limit. These data are, to date, the most persuasive demonstration of CdGM states in the quantum limit but it should be noted that similar phenomenology was also reported in ref. 7.

Results and Validity:

Higher order CdGM states were only observed in one of the reported vortices. I think the manuscript would benefit from further examples of this type of vortex in the supplementary information to evidence the repeatability and universality of these phenomena.

Aside from this, the data are of good quality and certainly evidence discrete sub-gap peaks near the center of vortices which are not inconsistent with CdGM states.

While not inconsistent with CdGM states the manuscript does not make a compelling argument for why these must be CdGM states and can not be attributed to other causes e.g field induced competing order or impurity states of defects pinning vortices.

It would be helpful if the two gaps referred to in figure 1c were labeled by arrows.

The manuscript describes a procedure for finding the bias voltage offset by finding the voltage at which there was zero current. It would be helpful if more detail was given on this procedure. Particularly, was the offset of the current to voltage amplifier calibrated or nulled? If not then this offset on the amplifier will lead to an error in the measured zero-bias point.

Presentation and Clarity:

The "waterfall" plots of spectra in figures 2 and 4 are very dense and it is difficult to discern the salient features without very close inspection. Fewer spectra and labeling of the important features of the spectra would be helpful.

Reviewer #3 (Remarks to the Author):

The study by Chen and coworkers reports on a study of the vortex-core bound states in FeTeSe. They study a crystal which is free of excess iron, and observe a very clean gap as well as well defined vortex core bound states. The vortex core bound states, similar to what is found in other iron-based superconductors, are asymmetric with respect to the Fermi level, and can be related to Caroli-Matricon-de Gennes (CMDG) states. The interpretation of the vortex core bound states in terms of being, at least for some bands, in the quantum limit is not entirely new and has been discussed previously, e.g. in ref. 11, the main novelty arises from the observation of bound states for $\mu > 1/2$. The paper is well written and the results presented in a clear manner. In some places, technical details are contained in the main manuscript which could be moved to the supplementary material, while at the same time the physics could be discussed in a bit more detail in the main text. The authors mention that the Fermi energy (or energy of band edge) extracted for the band which causes the CMDG states from their data is consistent with that obtained from ARPES for some of the bands, but it would be good to quote the relevant Fermi energies which have been determined from ARPES to spare the reader to have to go to these references.

Did the authors spatially map the vortex core bound states for $\mu = 1/2, 3/2$ and $5/2$? It would be very interesting to see whether they have internal spatial structure.

It is a bit puzzling to see that the vortex core bound states vary as much as they do between different vortices. Do the authors have any evidence from spectra in zero field in the same location whether there is anything special about one of the two cases (i.e. additional bound states or not?)

I would be happy to see this paper published in Nature Communications after the authors have considered the minor revisions mentioned in my report.

Detailed comments:

* Typo in title, the states are called "Caroli-de Gennes-Matricon" (Matricon without r before t – see reference 1), there are a few more typos throughout the text which should be fixed

* How does the composition of the crystal measured by EDX or seen by STM (extracted as, e.g., in Sci Adv 1, e1500206) compare with the nominal composition?

* I suggest moving the discussion of the bias offsets to the supplementary. It seems a bit risky to use the criterion of zero current to define the bias offset, given that there is a superconducting gap. Also using zero current to calibrate the bias offset only works if there is no current offset, so it would be good if the authors could elaborate a bit on this when moving the discussion to the supplementary.

* I find the discussion of zero energy bound states in ref. 28 rather unhelpful, as the experiment is a different one. The authors don't find evidence for them in vortex cores, and that's as much as they can say. They don't probe states at excess iron atoms, which is what ref. 28 does.

* Fig. S3b refers to a line 4 in fig. 2 which is not shown there, but probably it is meant to refer to fig. S2a

Reply to Reviewers' comments

Words in blue colour are our responses

Reviewer #1 (Remarks to the Author):

=====

The manuscript presents new detailed low temperature STM data revealing the existence of Caroli-de Gennes-Matricon states in tunneling spectra in FeTe_{0.55}Se_{0.45} superconductors. These data shed new light on the nature of quantum states in the vortex core in superconductors and will be certainly of great interest to the readers of Nature Communications and other Journals.

The data are in general well presented and the manuscript can be accepted for publication in Nature Communications, provided that the authors revise the part describing the asymmetry of the CdGM peaks in tunneling spectra. More specifically, more convincing answers should be given concerning the possible nature of the asymmetry of the observed CdGM peaks with respect to zero bias.

Response: We thank this referee for a general appreciation to our work. Concerning the description on asymmetry of the CdGM peaks, we made two modifications. Firstly, we revised the description in a more clear way, see the added words " ...which manifests in two ways. (1) Three bound state peaks appear in the positive energy side, but only one in the negative energy side; (2) The differential conductance intensity of the lowest bound state peak on negative energy is slightly higher than that on positive energy." Secondly, we specially stated out the Fermi energies of the related bands and added: " ARPES measurements²¹ show that the two hole bands (α_2 and α_3) with superconducting gaps of about 2 meV cross the Fermi level with $E_F \approx 4$ meV, and the electron band has a bottom energy of about 10 meV from the Fermi

level with a superconducting gap of about 5 meV. Since our measurements reveal mainly the maximum gap of about 2 meV, we attribute the CdGM bound states mainly to the collective contributions of the two hole bands. Due to the shallow hole bands, the bound state intensity and distributions are naturally asymmetric with respect to zero bias.”

Reviewer #2 (Remarks to the Author):

=====

Chen et al. report the observation of peaks in the tunneling conductance inside superconducting vortices in FeTeSe using scanning tunneling microscopy. They argue that the structure of these peaks both in space and energy is consistent with them being the result of discrete Caroli-de Gennes-Matricon (CdGM) states. Furthermore, they conclude that the observation of discrete CdGM states means that FeTeSe has a small Fermi energy.

Novelty, Significance and Interest:

The main novel claim of this manuscript is the observation of discrete CdGM states in superconducting vortices, an experimentally demanding task. The observation of discrete CdGM states implies a small Fermi energy in FeSeTe. This has previously been inferred using ARPES in refs. 21 and 22 and so will not significantly influence thinking in the field. Aside from this the measurements presented do not shed light on the physics of FeTeSe and related compounds. Consequently, it is mainly of interest to STM experimentalists and theorists who work on vortex core structure.

Response: We appreciate the critical reading and constructive comments of this referee. While we believe our paper sheds new lights in several aspects. Indeed, the small Fermi energy has been inferred from several experiments in iron based superconductors, while the *discrete* CdGM bound states due to the small Fermi energy have never been reported previously in iron based

superconductors. It is thus very important to directly observe such discrete bound state peaks corresponding to different energy levels, especially those of higher orders. Now our work has significance in following aspects: (1) direct observation of the discrete CdGM bound states in the quantum limit, and the main novelty of our paper arises from the observation of bound states for $\mu > 1/2$, as appreciated by all referees; (2) to prove that the Fermi energy in FeSeTe is indeed very small from a different point of view and consequently leading to the observation of the discrete CdGM states; (3) the observed asymmetry of the CdGM peaks with respect to the zero-bias and different behaviors in different vortices will stimulate further theoretical efforts.

The authors present measurements of vortices exhibiting sub-gap states that fall into two categories a) discrete peaks symmetrically positioned around zero bias b) single peak slightly offset from zero bias. As stated in the manuscript, these two phenomena have previously been observed (albeit not together) in cuprates (refs. 8,9,13), YNi₂B₂C (ref. 7) and other iron-based superconductors (refs. 10-12).

One of the reported vortices exhibits multiple sub-gap peaks with spatial structure consistent with CdGM states in the quantum limit. These data are, to date, the most persuasive demonstration of CdGM states in the quantum limit but it should be noted that similar phenomenology was also reported in ref. 7.

Response: We agree with the referee that our results provide "... to date, the most persuasive demonstration of CdGM states in the quantum limit". Actually, by having a closer look at the results in Ref.7 in YNi₂B₂C, one finds that their results are more close to the second case mentioned above, namely "a main single peak slightly offset from zero bias". In addition to that, they found a tiny second peak on the positive bias, which was attributed to the 2nd discrete CdGM bound state peak. Concerning the vortex core in

cuprates, beside the rather symmetric two peaks around zero bias, so far no observations on the second or third peaks associating with the related energy levels are reported. These peaks may be induced by the competing orders. Therefore, our results have clear significance in proving the discrete CdGM states in the quantum limit.

Results and Validity:

Higher order CdGM states were only observed in one of the reported vortices. I think the manuscript would benefit from further examples of this type of vortex in the supplementary information to evidence the repeatability and universality of these phenomena.

Response: We highly appreciate this constructive suggestion. Indeed a firmly established scientific conclusion needs to be repeated. We follow this suggestion and did new rounds of measurements on other samples. It is not difficult to find the similar behavior in some vortices. In the newly added Supplementary Fig. 6 we present the data measured on other two vortices. They also show the discrete energy levels (up to third level) at positive bias, although the oscillating amplitude is not as strong as that presented in Fig.2. We fully believe they have the same origin.

Aside from this, the data are of good quality and certainly evidence discrete sub-gap peaks near the center of vortices which are not inconsistent with CdGM states.

Response: We appreciate the precise judgement and positive comments on our work by this referee. Indeed, combining the repeated results of control experiment, we believe that our results reveal the evidence of discrete sub-gap peaks near the center of vortices, which is the very merit of our work.

While not inconsistent with CdGM states the manuscript does not make a

compelling argument for why these must be CdGM states and can not be attributed to other causes e.g field induced competing order or impurity states of defects pinning vortices.

Response: This is again a pertinent comment, while we can easily argue that the discrete bound state peaks observed here are not due to the competing orders or impurity states of defects, but arising from the CdGM states in the quantum limit. We support this point with the following arguments: (1) We observed three bound state peaks at the positive bias side within the vortex core. For competing orders or impurity states of defects, if existing, it is difficult to imagine why we should have three characteristic energy scales. (2) The energies of these peaks from the vortex shown in Fig.2 are +0.45, +1.2, and +1.8 meV, which gives a ratio of 1:2.7:4, this is not so far from the ideal case of 1:3:5 of the CdGM 1st, 2nd and 3rd energy levels. This consistency cannot be excluded by simply arguing that it is induced by accident. (3) In iron based superconductors, the possible competing orders include the nematic phase and the antiferromagnetic order. In FeTe_{1-x}Se_x on the doping level of $x=0.45$, it is known that these orders are absent. Given that these competing phases may emerge in the vortex core, but the energy scales should be much larger than those observed here. In addition, we have measured the spatial evolution of tunnelling spectra without magnetic field in the same region where the vortex would appear under a magnetic field. It is found that the tunnelling spectra are rather uniform without impurities or defects in the vortex core region. Related discussions have been added in the revised version.

It would be helpful if the two gaps referred to in figure 1c were labeled by arrows.

Response: We thank the referee for this suggestion. As mentioned in the paper, for the spectra measured at zero external magnetic field, there are usually one or two pairs of coherence peaks with peak energies ranging from

about ± 1.1 mV to ± 2.1 mV. We added two arrows on one typical spectrum to mark the two energy gaps.

The manuscript describes a procedure for finding the bias voltage offset by finding the voltage at which there was zero current. It would be helpful if more detail was given on this procedure. Particularly, was the offset of the current to voltage amplifier calibrated or nulled? If not then this offset on the amplifier will lead to an error in the measured zero-bias point.

Response: We thank the referee for reminding us about the issue of current offset. We have added more detailed information about how to find the bias voltage offset in the revised version. While following the suggestion of the third referee, we move the detailed description of this art to Supplementary Information, together with a newly added figure (Supplementary Fig. 8). In fact, before the bias-offset calibration, the tunneling current offset is already nulled in the tip-retraction process with a tip-sample distance of about 100 nm. Then we move the tip approaching the sample, start the measurement and voltage calibration.

Presentation and Clarity:

The “waterfall” plots of spectra in figures 2 and 4 are very dense and it is difficult to discern the salient features without very close inspection. Fewer spectra and labeling of the important features of the spectra would be helpful.

Response: Thanks for this suggestion, and this has been fixed in the new version. Since now the presentation in Fig.2 is only done for half of the total number of spectra, we still present the original “waterfall” plots of dense spectra in Supplementary Fig. 2.

Reviewer #3 (Remarks to the Author):

=====

The study by Chen and coworkers reports on a study of the vortex-core bound states in FeTeSe. They study a crystal which is free of excess iron, and observe a very clean gap as well as well defined vortex core bound states. The vortex core bound states, similar to what is found in other iron-based superconductors, are asymmetric with respect to the Fermi level, and can be related to Caroli-Matricon-de Gennes (CMDG) states. The interpretation of the vortex core bound states in terms of being, at least for some bands, in the quantum limit is not entirely new and has been discussed previously, e.g. in ref. 11, the main novelty arises from the observation of bound states for $\mu > 1/2$.

The paper is well written and the results presented in a clear manner. In some places, technical details are contained in the main manuscript which could be moved to the supplementary material, while at the same time the physics could be discussed in a bit more detail in the main text. The authors mention that the Fermi energy (or energy of band edge) extracted for the band which causes the CMDG states from their data is consistent with that obtained from ARPES for some of the bands, but it would be good to quote the relevant Fermi energies which have been determined from ARPES to spare the reader to have to go to these references.

Response: We thank the referee for the careful review and positive comments, especially the precise judgement of the merit of our paper. We follow the referee's suggestion, and add the values of Fermi energies determined from ARPES to the text. Some technical details, such as the calibration of zero bias voltage etc., have been moved to the Supplementary part.

Did the authors spatially map the vortex core bound states for $\mu = 1/2, 3/2$ and $5/2$? It would be very interesting to see whether they have internal

spatial structure.

Response: We thank the referee for this interesting and constructive suggestion. While by mapping out the dI/dV at the three particular energies, we did not observe clear fine structures. We attribute this to two major reasons: (1) The three peaks are locating on the segment of spectrum with a high slope, which greatly diminishes the contrast of the oscillating behavior. Thus a 2D mapping cannot show the expected oscillations. (2) The radius of the vortex core (coherence length) here is quite small, which is about 2.5 nm. According to the theoretical understanding to the CdGM states, the periodicity of the oscillation is about $1/k_F$, for the two related hole bands, k_F is about 0.7 nm^{-1} . In this case, only 1-2 periodicity of the oscillation can be observed. Taking the fact of exponential decaying of the peak intensity, it is really hard to see this oscillation. Actually some traces of such oscillation can be seen in Supplementary Fig. 4.

It is a bit puzzling to see that the vortex core bound states vary as much as they do between different vortices. Do the authors have any evidence from spectra in zero field in the same location whether there is anything special about one of the two cases (i.e. additional bound states or not?)

Response: We appreciate this question and it is a natural concern. Indeed, the vortex core bound states behave differently in different vortices. We have repeated to observe more vortices with three bound state peaks on the positive bias in different samples, which is added as a new figure of Supplementary Fig. 6. For these newly observed vortices, the spatial evolution of spectra at zero field are also measured and shown in the same figures. One can see that the spectra look quite uniform. However, we must emphasize that most vortices show the core states with a strong peak locating at a slightly positive bias. We have intentionally measured the spatial evolution of spectra at zero field in the area where one such vortex would appear when a magnetic field is applied, shown in Supplementary Fig.

7. It is easy to see that in the core region the gap becomes clearly smaller than in the region far away. Therefore, it is tempting to conclude that the defect induced pinning can modify the structure of the vortex core bound states and may smear up the peaks of different quantum levels.

I would be happy to see this paper published in Nature Communications after the authors have considered the minor revisions mentioned in my report.

Detailed comments:

* Typo in title, the states are called "Caroli-de Gennes-Matricon" (Matricon without r before t – see reference 1), there are a few more typos throughout the text which should be fixed

Response: We thank the referee for pointing out this negligence. This has been fixed. The linguistic problems have been corrected as much as we can by rephrasing the text in many places.

* How does the composition of the crystal measured by EDX or seen by STM (extracted as, e.g., in Sci Adv 1, e1500206) compare with the nominal composition?

Response: The composition of the crystal is nominal one as mentioned in the Method part. We add such information in the main text in the revised version. Following the referee's suggestion, we also do the counting to the bright and dark surface atoms in Fig. 1a, and the numbers of them are roughly 730 over 611, respectively. Then, we can calculate the composition from the surface as $\text{FeTe}_{0.54}\text{Se}_{0.46}$, which is close to the nominal formula $\text{FeTe}_{0.55}\text{Se}_{0.45}$. We also add these discussions to the main text.

* I suggest moving the discussion of the bias offsets to the supplementary. It seems a bit risky to use the criterion of zero current to define the bias offset, given that there is a superconducting gap. Also using zero current to

calibrate the bias offset only works if there is no current offset, so it would be good if the authors could elaborate a bit on this when moving the discussion to the supplementary.

Response: We thank the referee for reminding us about the calibration of current, which are indeed neglected in the old version. Following the suggestion of this referee, we move the detailed description to Supplementary Information, together with a newly added figure (Supplementary Fig. 8). We have added more detailed information about how to find the bias voltage offset in the revised version. In fact, before the bias-offset calibration, the tunneling current offset is already nulled in the tip-retraction process with a tip-sample distance of about 100 nm. Then we move the tip approaching the sample, start the measurement and voltage calibration.

* I find the discussion of zero energy bound states in ref. 28 rather unhelpful, as the experiment is a different one. The authors don't find evidence for them in vortex cores, and that's as much as they can say. They don't probe states at excess iron atoms, which is what ref. 28 does.

Response: We agree and thus soften the tone in the description. Indeed what we want to say is that we did not find evidence of the bound state peaks at zero bias. We removed the sentences reminding any links between their observation and the possible off-set problem.

* Fig. S3b refers to a line 4 in fig. 2 which is not shown there, but probably it is meant to refer to fig. S2a

Response: Thanks for this suggestion. This has been fixed.

REVIEWERS' COMMENTS:

Reviewer #2 (Remarks to the Author):

In response to my scientific and technical concerns the authors have replied comprehensively and edited their manuscript appropriately so as to bolster and clarify their methods and claims.

As to the impact and significance, both mine and the authors' comments speak for themselves and require no further clarification on my part.

Reviewer #3 (Remarks to the Author):

The authors have accounted for my comments, and I am happy for the manuscript to be accepted for publication as is.

Reply to Reviewers' comments

Words in blue colour are our responses

Reviewer #2 (Remarks to the Author):

=====

In response to my scientific and technical concerns the authors have replied comprehensively and edited their manuscript appropriately so as to bolster and clarify their methods and claims.

As to the impact and significance, both mine and the authors' comments speak for themselves and require no further clarification on my part.

Response: We appreciate the careful review of this referee and thank the referee for the positive comments on our revisions/responses.

Reviewer #3 (Remarks to the Author):

=====

The authors have accounted for my comments, and I am happy for the manuscript to be accepted for publication as is.

Response: We thank the referee for the encouraging comments and the suggestion of "publication as is".